# Potential Biomarker and Therapeutic Tools for Pathological Cardiac Hypertrophy and Heart Failure: Extracellular Vesicles

**DOI:** 10.3390/ijms27010095

**Published:** 2025-12-22

**Authors:** Jinpeng Sun, Dongli Zhou, Min Cheng

**Affiliations:** 1Department of Cardiology, Union Hospital, Tongji Medical College, Huazhong University of Science and Technology, Wuhan 430022, China; sunsunjinjin@163.com (J.S.); zdl1020@hust.edu.cn (D.Z.); 2Hubei Key Laboratory of Biological Targeted Therapy, Union Hospital, Tongji Medical College, Huazhong University of Science and Technology, Wuhan 430022, China; 3Hubei Engineering Research Center of Immunological Diagnosis and Therapy for Cardiovascular Diseases, Union Hospital, Tongji Medical College, Huazhong University of Science and Technology, Wuhan 430022, China

**Keywords:** extracellular vesicles, exosomes, cardiac hypertrophy, heart failure, microRNA

## Abstract

Cardiovascular disease remains the leading global cause of death. Pathological cardiac hypertrophy is a key precursor to heart failure (HF), a condition with high morbidity and mortality. Extracellular vesicles (EVs) have emerged as crucial mediators of intercellular communication, carrying bioactive cargoes that reflect cellular state and influence recipient cell function. This review provides a focused and integrative perspective distinct from broader overviews, by dissecting the dynamic, cell-type-specific roles of EVs across the continuum from pathological hypertrophy to overt HF. We critically synthesize evidence on how EVs derived from cardiomyocytes, fibroblasts, immune cells, and adipocytes orchestrate maladaptive remodeling. Furthermore, we evaluate their dual utility as emerging diagnostic biomarkers and as engineerable therapeutic vectors. By highlighting recent advances in EV engineering for targeted delivery and discussing persistent translational challenges, this article offers a unique mechanistic-to-translational viewpoint aimed at advancing the therapeutic application of EVs in cardiovascular medicine.

## 1. Introduction

Heart failure (HF) represents the common end-stage of numerous cardiovascular diseases, characterized by a complex pathophysiology involving neurohormonal activation, inflammatory responses, and extensive myocardial remodeling [1]. Cardiac hypertrophy, initially a compensatory response to stress, becomes pathological and a major risk factor for HF when sustained [2]. Understanding the intercellular crosstalk that drives this detrimental transition is paramount. Beyond direct contact and soluble factors, extracellular vesicles (EVs) have been recognized as essential conduits for cell-to-cell communication, shuttling proteins, lipids, and nucleic acids to modulate recipient cell phenotype [3].

While the general roles of EVs in cardiovascular health and disease have been reviewed [4,5], there is a need for a focused analysis on their specific functions within the pathological hypertrophy-to-HF axis. This axis involves a coordinated yet dysregulated dialogue among cardiac cell types. Recent reviews have covered EVs in broad cardiovascular contexts [6], specific microRNA (miRNA) families [7], or general therapeutic potential [8]. This manuscript distinguishes itself by providing an integrated, cell-centric, and dynamic analysis of how EVs from defined cellular sources (cardiomyocytes, fibroblasts, macrophages, adipocytes) contribute to specific stages of disease progression. The study aims to (1) detail the biomarker potential of EVs in HF of varied etiologies; (2) elucidate the mechanistic networks of EV-mediated crosstalk in driving hypertrophy and failure; (3) critically appraise the latest strategies in engineering EVs for targeted therapy; and (4) discuss the translational hurdles and future directions. This focused approach offers novel insights into both the pathogenesis and potential precision treatment of cardiac hypertrophy and HF.

## 2. Biogenesis and Function of EVs and Exosomes

EVs are nanometer to micrometer-sized particles confined by lipid bilayer membranes, appearing to be released by all types of cells and present in various body fluids such as blood, urine, semen, and saliva [8]. According to synthesis and release mechanisms, EVs encompass various subtypes including exosomes, apoptotic blebs, and other EV subgroups [9].

In early studies, the term “EXOSOME” (50–100 nm in diameter) was first coined in 1981 by Trams et al. [10], then identified in 1983 by Pan B.-T. et al. from sheep reticular cells [11]. Their biogenesis starts from cell membrane invagination to form endosomes, the early sorting endosomes (ESEs) encapsulate a range of bioactive substances from the endoplasmic reticulum (ER) and Golgi networks as well as from the cytosol [12]. While ESEs mature into late sorting endosomes (LSEs), the sorting cargoes on the endosome membrane drive an inward invagination of the membrane to form intraluminal vesicles (ILVs, the future exosomes), which gives endosomes the multivesicular appearance, thus the name of multivesicular bodies (MVBs) [13]. The endosomal sorting complex required for transport (ESCRT) plays an important role in the process of exosome generation such as cargo loading, ILV formation, and exosome release into the extracellular space [12,14,15,16]. Nevertheless, evidence also suggests ESCRT-independent mechanisms [17] and contributions from other cellular components, such as lipid rafts [18] and four transmembrane domain proteins [19]. Notably, sorting and packing of lipids, proteins, RNAs including miRNAs, circular RNAs (circRNAs) and messenger RNAs (mRNAs) [20], and other metabolites into ILVs are a function of material exchange of endosomes with other subcellular organelles such as Trans-Golgi Network [21,22]. Evidence suggests these bioactive cargoes are selectively loaded into ILVs depending on the physiological or pathological conditions of the source cells. Additionally, whereas a portion of MVBs fuse with lysosomes for degradation [21,23,24], other MVBs are transported along cytoskeleton with the involvement of Ras-associated binding protein (Rab) family members, docking on the inner plasma membrane and releasing ILVs to extracellular space as exosomes [25,26] (Figure 1). Thus, exosomes carry complex bioactive cargoes that reflect the functional and metabolic states of their source cells. They can also deliver these cargoes to other organs and modulate the function of target cells [27,28]. These features, combined with their easy accessibility via various biofluids, make them ideal biomarkers for insights into various disease diagnosis, prognosis, and pathogenesis. Due to their multiple functions in transmitting information between cells, EVs play a critical role in immune responses, viral pathogenicity, cardiovascular diseases, central nervous system-related diseases, and cancer progression [29]. Traditional exosome isolation and purification methods mainly involve differential centrifugation, density gradient centrifugation, size exclusion chromatography, ultrafiltration, precipitation, and immunocapture assays [30]. Other new exosome isolation strategies including charge-based separation techniques, microfluidics, the synthetic peptide (Vn96)-based isolation method, EXODUS, chimeric nanocomposites-based technology, SAP-based technology, and cocktail strategy [31]. Due to the inherent problems of EVs, such as their heterogeneity in physicochemical properties, it is unclear and may derivatively influence the choice of different separation techniques. It is also hard to say which method is the best choice, as there are so many measurements that need to be measured, such as yield, purity, and separation efficiency.

In general, the International Society for Extracellular Vesicles (ISEV) recommends the use of the generic term “EV” and operational extensions of that term rather than the use of inconsistently defined and sometimes misleading terms such as “exosomes” and “ectosomes”, terms associated with difficult-to-define biogenesis pathways [8]. Thus, in this paper, we respect the original researchers’ definition of EVs or exosomes, and in the review discourse, we use EVs collectively.

## 3. Cardiac Hypertrophy and HF

Cardiac hypertrophy, characterized by the hypertrophic growth of cardiomyocytes, represents a fundamental adaptive response of the heart. It is controlled by sophisticated neurohumoral factors, cellular metabolism, and intracellular signals [32]. Myocardial hypertrophy can be physiological, as seen in response to pregnancy and physical exercise, leading to increased myocyte size, ventricular diastolic compliance, and contractility [33,34]. In contrast, pathological hypertrophy is driven by sustained stressors as pathological pressure and/or volume overload and functional decompensation of conditions such as myocardial infarction and hypertension [2]. This maladaptive process is characterized not only by enlarged cardiomyocytes but also by interstitial fibrosis, electrophysiological abnormalities, and vascular insufficiency, which collectively lead to systolic and diastolic dysfunction, ventricular dilation, and ultimately HF [2,35,36].

Cardiac hypertrophy is also a major feature of hypertrophic cardiomyopathy (HCM) [37]. HCM is typically a monogenic disease with autosomal dominant inheritance, most commonly caused by variants in the *MYBPC3* or *MYH7* genes encoding key components of the contractile machinery [38]. These genetic defects lead to myofibrillar disarray, cardiomyocyte hypertrophy, and fibrosis, which impair diastolic function and predispose individuals to HF or sudden cardiac death [39]. Clinically, HCM is diagnosed when septal or left ventricular wall thickness reaches ≥ 15 mm (or ≥13 mm in individuals with a positive family history) on echocardiography [40]. Based on the left ventricular outflow tract gradient (LVOTG), HCM is classified as obstructive (labile or basal) or nonobstructive. It occurs in approximately 1 in 200 to 1 in 500 people around the world, and although it can be associated with a completely normal life, it can also cause significant issues with symptoms and even sudden cardiac death, including being perhaps the most common cause of death in young athletes [41,42].

Cardiac pathological hypertrophy often leads to progressive loss of heart function. The declining cardiac output struggles to meet the body’s metabolic and oxygen needs, eventually leading to organ failure and death [2,35,36]. Beyond imaging and functional assessments, biomarkers like B-type natriuretic peptide (BNP) are crucial for diagnosing and stratifying HF [43]. According to the latest 2022 AHA/ACC/HFSA Guidelines, HF is further classified into three categories based on left ventricular ejection fraction (LVEF), as heart failure with reduced ejection fraction (HFrEF, LVEF < 40%), heart failure with mid-range ejection fraction (HFmrEF, LVEF 41–49%), and heart failure with preserved ejection fraction (HFpEF, LVEF ≥ 50%). The pathophysiology involves a deregulated neurohormonal axis (renin-angiotensin-aldosterone and beta-adrenergic systems) and metabolic dysfunction, which are targeted by current therapies including ACE inhibitors, ARBs, aldosterone antagonists, beta-blockers, and SGLT2 inhibitors [44]. Despite these treatments, HF remains a leading cause of global mortality, underscoring the critical need for early intervention at the stage of cardiac hypertrophy.

While the roles of direct cell–cell contact, paracrine factors, and cell–extracellular matrix interactions in hypertrophy are well-established [45], the significance of EVs, including exosomes in this pathogenic communication, is an emerging and less understood frontier. Notably, EVs are gaining attention for their potential role across the HF spectrum; for instance, circulating miR-133a levels are significantly reduced in patients with HF, suggesting a link to cardiomyocyte apoptosis [46,47]. This review summarizes recent discoveries concerning the roles and molecular mechanisms by which EVs and their cargo regulate the pathogenesis of cardiac hypertrophy and HF, and explores their associated therapeutic implications.

## 4. Circulating EVs as Biomarkers for Cardiac Hypertrophy and HF

EVs are detected in all bodily fluids, carrying diverse bioactive molecules that reflect the identities and pathophysiological states of their source tissues. At the same time, cargoes in EVs are well protected from tissue environment by the enclosing bilayer membrane [48,49,50], as ideal sources of biomarkers that can aid in disease diagnosis, prognosis, and therapy. For example, in the peripheral blood, ~10–15% of the total circulating miRNAs are carried within EVs [51,52,53,54], and the molecular signature of the cardiomyocyte-derived EVs may help aid the evaluation of cardiomyocytes’ functional and metabolic states and potential development of HF.

It has been shown that miR-486 and miR-146a, two cardiomyocyte-enriched miRNAs, regulate cell survival and attenuate inflammation, respectively [55,56,57]. Beg et al. measured miR-146a, miR-486, and miR-16 in the plasma and plasma exosomes of 40 HF patients (ischemic and non-ischemic) and 20 healthy individuals. Interestingly, they found that miR-146a/miR-16 and miR-486/miR-16 ratios were significantly elevated in the plasma exosomes, but not in the total plasma of HF patients [58]. However, the sample size of this study was small, no stratification by HF subtype was performed, and differential centrifugation was used for exosome isolation, which may lead to lipoprotein contamination and affect the specificity of the results. Future large-scale, multicenter studies using more precise isolation techniques (e.g., immunocapture) are needed to validate the diagnostic value of these miRNA ratios.

HF is classified as HFrEF, HFmrEF, and HFpEF, which result from different etiology and require different treatment strategies. Tao Wu et al. analyzed three exosomal miRNAs, miR-92b-5p, -192-5p, and -320a in the sera of patients with HFrEF patients hospitalized for acute HF (AHF), including de novo AHF and acute decompensated HF, and 30 healthy volunteers. The authors found that serum exosomal miR-92b-5p was elevated in the acute HFrEF patients [59].

An important feature of cardiac hypertrophy is fibrosis, which contribute to the pathogenesis of arrhythmia and HF [60,61]. Wang et al. examined a panel of nine candidate miRNAs (miRs-221, 15a, 122, 21, 29, 30d, 133a, 425, and 744) in the plasma exosomes of 31 AHF patients and paired control subjects. They observed a significantly higher miR-21 level and significantly lower miR-425 and miR-744 levels in the AHF exosomes. The authors found that HF stresses blunt miR-425 and miR-744 induction in cardiac fibroblasts (CFs), which is accompanied with upregulated TGFβ1, collagen 1, and α-SMA expression and fibrogenesis. Thus, a reduction in the level of fibroblasts-derived exosomal miR-425 and miR-744 may predict cardiac fibrosis and HF [62].

### 4.1. Acute Myocardial Infarction Associated HF

Matsumoto et al. screened a panel of 377 miRNAs in the sera collected from acute myocardial infarction (AMI) patients at a median of 18 days after the onset of AMI, then correlated the miRNA levels with the disease states in 21 patients, who developed ischemic HF in one year post-AMI, and in 65 patients, who experienced no subsequent cardiovascular events after discharge. The authors identified three p53-responsive miRNAs, miR-192, miR-194, and miR-34a, whose concentrations in the serum exosomes were significantly correlated with the later occurrence of ischemic HF. Thus miR-192, miR-194, and miR-34a may function as circulating regulators of HF development via p53 pathway. In support, the authors found that in cultured cardiomyocytes, knockdown of the three miRNAs improved, while their overexpression attenuated, cell survival with doxorubicin treatment [63]. Thus, elevated levels of miR-192, miR-194, and miR-34a in the circulating exosomes at early convalescent stages of AMI may indicate the involvement of the p53 pathway and predict the risk of HF development in AMI patients. This study is the first to link p53 pathway-related miRNAs to the risk of HF after AMI, but it has notable limitations: only 21 HF patients were included, the impact of treatments such as statins on miRNA levels was not considered, and the consistency of these three miRNAs’ expression in HF of other etiologies remains unclear, limiting their application as broad-spectrum biomarkers.

Vascular endothelial dysfunction is a major contributor to not only coronary artery disease (CAD) but also HF development. Nozaki et al. measured endothelial cell (EC)-derived EVs in the plasma of a cohort of 169 patients with NYHA class I or higher HF. They found that the levels of EC-derived EVs increased significantly with NYHA functional class [64]. Similarly, Berezin et al. found that EC-derived apoptotic microparticles (CD144^+^/CD31^+^/Annexin V^+^ EVs and CD31^+^/Annexin V^+^ EVs) predict poor prognosis in HF patients, and that CD31^+^/Annexin V^+^ EVs predict all-cause mortality in these patients. Thus, the levels of EC-derived EVs and apoptotic microparticles may help in identifying HF patients with poor outcome [65,66].

### 4.2. Peripartum Cardiomyopathy Associated HF

Peripartum cardiomyopathy (PPCM)-associated HF (PPCM-HF) has a high mortality rate but there is a lack of diagnostic and prognostic biomarkers. Halkein compared levels of peripheral blood miRNAs in 30 dilated cardiomyopathy patients and 38 PPCM patients. They found that circulating exosomal miR-146a was explicitly greater in the PPCM patients. Furthermore, the circulating exosomal miR-146a in the acute PPCM patients declined with standard HF treatment, making miR-146 a potential biomarker specifically for PPCM-associated HF [67]. Subsequently, J. G. et al. further demonstrated that EVs from miR-146a-loaded endotheliocytes suppress SUMO1 expression and induce cardiac dysfunction in maladaptive hypertrophy [68]. Furthermore, Vaughan O. R. et al. found that umbilical cord serum from women with obesity upregulates pathological hypertrophy genes (atrial natriuretic factor, brain natriuretic peptide) in neonatal rat cardiomyocytes, an effect abolished by physical/enzymatic disruption. Although EVs from this serum showed elevated miR-142/17, transfection of these miRNAs did not induce hypertrophy. The placental-trophoblast-conditioned medium only increased atrial natriuretic factor [69]. These findings indicate that not specific EV-miRNAs in maternal circulation and placenta, induce pro-hypertrophic gene programming and may contribute to offspring cardiac dysfunction.

### 4.3. Myxomatous Mitral Valve Disease-Associated HF

Human mitral valve prolapse (HVP) is associated with aging and can also lead to the development of HF. Dog myxomatous mitral valve disease (MMVD) displays a pathology close to HVP. Yang V. K. et al. compared exosomal miRNAs and total miRNAs in the dog MMVD, MMVD with chronic HF (MMVD-CHF), and dog aging. They observed changes in exosomal miRNA in dogs as they age (miR-9, miR-495 and miR-599), develop MMVD (miR-9 and miR-599), and progress from MMVD to chronic heart failure (CHF) (miR-181c and miR-495), whereas none of these changes was statistically significant in total plasma (the false discovery rate was set < 15%) [70]. Thus, compared to total plasma miRNAs, exosomal miRNAs (miR-181c and miR-495) seem to be more specific diagnostic biomarkers for MMVD-CHF.

Collectively, these studies have identified that several miRNAs carried in the circulating exosomes may be used as biomarkers for HF and associated pathologies. Nevertheless, the changes of these miRNAs do not seem to be well conserved across different studies. While this may be attributed to many potential differences such as the study subjects, the stage and severity of disease, medications, time, and methods of sample collections and processing, there is an urgent need for large-sized multicenter clinical studies. Importantly, circulating exosomes are a mixture of exosomes derived from different tissues and organs with unknown proportions, attainment, and characterization of cell-type specific exosomes in the circulation, and will significantly facilitate the identification of miRNAs and mechanisms of exosomal miRNA-mediated inter-organ crosstalk in the development of HF. These findings exemplify possible roles of EVs as diagnostic markers. However, the accuracy of a given EV-based test mandates careful validation.

### 4.4. Diabetic Cardiomyopathy-Associated HF

Diabetic cardiomyopathy (DCM) is a major complication of type 2 diabetes, characterized by cardiac hypertrophy, fibrosis, and diastolic dysfunction [71]. A key pathological feature contributing to this dysfunction is impaired angiogenesis in the diabetic heart. Wang et al. elucidated diabetic cardiomyocytes secreting exosomes enriched with miR-320, which, upon transfer to cardiac endothelial cells, suppress angiogenesis by downregulating key pro-angiogenic targets such as *IGF-1* and *Hsp20*. This exosome-mediated miR-320 delivery establishes a direct paracrine mechanism by which diabetic cardiomyocytes actively inhibit vascular repair, thereby exacerbating myocardial injury [72]. A separate study using an STZ-induced diabetic HFpEF rat model demonstrated that the consistent downregulation of exosomal miR-30d-5p and miR-126a-5p correlated with cardiac dysfunction, highlighting their potential as diagnostic biomarkers [73]. A marked decrease in plasma miR-21 levels has been observed in T2DM patients with DCM relative to those without, with this differential expression showing high diagnostic value (AUC = 0.899), that is greater than that of HbA1c [74]. Collectively, these findings underscore the dual functionality of exosomal miRNAs in diabetic heart disease, acting as both pathogenic mediators of cellular injury and promising circulating biomarkers.

### 4.5. Hypertrophic Cardiomyopathy (HCM)-Associated HF

Hypertrophic cardiomyopathy (HCM) is characterized by primary myocardial hypertrophy and often progresses to HF. James V. et al. demonstrated that exosomal RNA cargo could serve as diagnostic biomarkers for HCM. By using a patient-specific hiPSC model carrying an *ACTC1* mutation, researchers found that differential expression of 12 snoRNAs was observed in EVs derived from HCM-induced pluripotent stem cell (hiPSC)-cardiomyocytes compared to those from control (WT hiPSC-cardiomyocyte derived) EVs. These included ten SNORDs and two SNORAs. Additionally, HCM hiPSC-CM EVs showed an increase in transcripts associated with epidermal growth factor and fibroblast growth factor signaling, 5-HT receptor-mediated signaling, angiotensin receptor signaling, β1- and β2-adrenergic signaling, and G protein-coupled receptor signaling [75]. Additionally, Zhang Y. et al. demonstrated that RNA-sequencing of brown adipose tissue (BAT) from iron-overloaded mice revealed downregulated genes significantly enriched in pathways related to hypertrophic cardiomyopathy. Furthermore, exosomes derived from iron-treated brown adipocytes were internalized by cardiomyocytes and induced their dysfunction, mechanistically linking iron-overloaded BAT to the onset of cardiomyopathy [76]. While these studies demonstrate the potential of exosomes and their cargo from various cellular origins (cardiomyocytes, adipocytes) as rich sources of mechanistic insight and biomarker candidates in HCM, the precise causal links between specific EV signatures and the progression to overt heart failure remain to be fully elucidated, highlighting a critical area for future research.

### 4.6. EVs in Other Etiologies of HF

Beyond the etiologies discussed above, emerging evidence suggests EV involvement in other causes of HF. For instance, a study demonstrates that M2 macrophage-derived exosomes alleviate viral myocarditis by delivering the long non-coding RNA AK083884, which targets the PKM2/HIF-1α axis to regulate macrophage metabolic reprogramming and M2 polarization [77]. The other study modeled Fabry disease cardiomyopathy through CRISPR/Cas9-mediated GLA knockout in human embryonic stem cells, which provides mechanistic insights relevant to pathological hypertrophy. Cardiomyocytes differentiated from these stem cells (GLA-null CMs) recapitulated disease hallmarks, including globotriaosylceramide (Gb3) accumulation and cellular hypertrophy. Proteomic analysis revealed a significant downregulation of Rab GTPases, which are crucial for exocytotic vesicle release. This disruption impaired autophagic flux and protein turnover, leading to increased reactive oxygen species and apoptosis, thereby delineating a vesicle trafficking-dependent pathway that connects metabolic storage to cardiomyocyte dysfunction and remodeling [78]. The study provided new insights for developing novel therapies for Fabry disease cardiomyopathy by targeting exosomal vesicle transport associated with Rab GTPase signaling. The role of EVs in infiltrative diseases like cardiac amyloidosis remains largely unexplored. Investigating EV profiles in these distinct etiologies could yield highly specific biomarkers and elucidate novel pathogenic mechanisms.

Collectively, clinical and translational studies have identified several miRNAs carried in circulating exosomes as candidate biomarkers for HF and related pathologies. However, the reported changes in these miRNAs are not well conserved across studies, likely owing to differences in study populations, HF etiology and stage, disease severity, concomitant medications, as well as the timing and methods of sample collection, EV isolation, and data normalization. Most of the available evidence still comes from relatively small, single-center, and methodologically heterogeneous cohorts, underscoring the urgent need for large, well-designed multicenter clinical studies. In addition, circulating exosomes represent a complex mixture released from multiple tissues and organs in unknown proportions; attaining and characterizing cell type-specific exosomes in the circulation will greatly facilitate the identification of disease-relevant miRNAs and the elucidation of exosomal miRNA-mediated inter-organ crosstalk in the development of HF. Overall, these findings exemplify the potential roles of EVs as diagnostic, phenotyping, and risk-stratification markers, and even as tools for treatment monitoring or EV-mediated immunomodulation in HF, but the accuracy and robustness of any given EV-based test will require rigorous analytical and clinical validation. The key clinical and translational studies discussed in this review are summarized in Table 1.

## 5. Role of EVs in Cardiac Hypertrophy and HF

In response to hypertrophic stress, the heart undergoes extensive remodeling, characterized by myocyte enlargement and interstitial fibrosis [80]. Prolonged stimuli lead to increased myocardial oxygen consumption and reduced myocardial compliance and contractility, resulting in decompensated cardiac hypertrophy, HF, or sudden cardiac death [81,82]. In this context, cardiac cells, including cardiomyocytes, fibroblasts, vascular endothelial cells (ECs), and macrophages have been shown to influence cardiomyocytes to activation of hypertrophy-associated signaling, including Ca^2+^ and Ca^2+^-dependent signaling, mitogen-activated protein kinase (MAPK), JAK-STAT, AMPK, and Wnt pathways [83,84,85,86,87]. EVs exert key functions for the intercellular communication of proximal and distant cells including cardiac cells and have been involved in the regulation of cardiomyocyte hypertrophy and HF [35] (Figure 2).

### 5.1. EVs Mediate Cardiac Hypertrophy Under Pathological Hypertrophic Stresses

Hypertension leads to pressure overload, and thus is a cause of cardiac hypertrophy. Jingwei Yu et al. injected C57BL/6 mice intravenously with exosomes isolated from spontaneously hypertensive rats (SHR). After 8 weeks, the mice displayed a thickened left ventricular wall and reduced cardiac function. Interestingly, the authors found that SHR’s serum exosomes carry angiotensinogen, renin, and angiotensin-converting enzyme (ACE), which are transported into cardiomyocytes to induce autocrine secretion of Ang II [88]. This study provides a novel mechanism by which pressure overload induces myocardial hypertrophy.

Uremic cardiomyopathy (UCM) is a common complication in patients with chronic kidney disease (CKD) and an important risk factor for death [89]. Cardiac hypertrophy is a major feature of UCM [90]. Chen K. et al. found that in experimental model of CKD, the expression of miR-205-5p and miR-208b-3p was significantly upregulated, while that of miR-215, miR-150, and miR-26b-5p was downregulated. Further mechanistic analysis revealed that the differentially expressed mRNAs were enriched in the activation of pro-hypertrophic signaling pathways such as rat sarcoma virus (Ras) and phosphatidylinositol 3-kinase/protein inase B (PI3K/Akt), accompanied by the suppression of metabolic pathways including Peroxisome Proliferator-Activated Receptor γ (PPARα) [91]. Interestingly, the other research has revealed a protective role of exosomal miR-27a-5p against this hypertrophy, it alleviates cardiomyocyte hypertrophy and cardiac dysfunction induced by indoxyl sulfate by directly targeting the USF2/FUT8 signaling axis [92]. The pathogenesis of UCM involves complex cardiorenal crosstalk. Pro-hypertrophic miRNAs (e.g., miR-205-5p) from pressure-overload models may contribute to UCM, while exosome-mediated protective signals (e.g., miR-27a-5p) may counteract uremic toxin damage. Future research should validate these miRNAs in unified UCM models, focusing on their roles as exosomal cargo in inter-organ signaling.

### 5.2. EVs Regulate Cardiac Inflammation and Hypertrophy

Inflammatory microenvironment plays an important role in the development of cardiac hypertrophy [2], and EVs have emerged as a potent regulator of tissue inflammation [93]. Hui Yu et al. found that Ang II-induced hypertrophic cardiomyocytes release more exosomes than control-treated cardiomyocytes. Treatment of mouse RAW264.7 macrophages with hypertrophic cardiomyocytes-derived exosomes triggered the secretion of pro-inflammatory cytokines IL-6 and IL-8. The authors further revealed that miR-155 plays a critical role in the initiation of inflammation in macrophages, and that hypertrophiccardiomyocyte exosomes induce phosphorylation of extracellular signal-regulated kinase (ERK), c-Jun N-terminal kinase (JNK), and p38 via miR-155. The authors concluded that exosomal microRNAs are important regulators of the inflammatory response in cardiac hypertrophy and may be a novel therapeutic target [94]. In another study, Hobuß L. et al. found that cardiac ischemia–reperfusion injury induced a massively increased pro-inflammatory cytokines in the heart of H19 long non-coding RNA (lncRNA) knockout mice, which was associated with exaggerated cardiac remodeling and hypertrophy. The authors further revealed that H19-dependent changes in cardiomyocyte-derived EV release and NF-κB signaling were evident [95]. In vitro, fibroblasts stimulated with TNF-α release EVs containing miR-27a, miR-28-3p, and miR-34a, which dysregulate the nuclear factor erythroid 2–related factor 2 (Nrf2) pathway that ordinarily prevents oxidative injury and protects against adverse cardiac remodeling and dysfunction [96]. Thus, exosomes serve to deliver multi-types of important pro-inflammatory mediators that promote cardiac hypertrophy. The pro-inflammatory role of EV-miR-155 is well demonstrated, yet the net effect of EVs on cardiac inflammation is context dependent. Some stem cell-derived EVs exert anti-inflammatory effects, underscoring the functional heterogeneity of EVs based on their cellular origin and activation state.

### 5.3. EVs Regulate Reactive Oxygen Species, Oxidative Stress, and Cardiac Hypertrophy

Reactive oxygen species (ROS) and oxidative stress play an important role in the pathogenesis of myocardial dysfunction, hypertrophy, fibrosis, and HF. In particular, miR-27a, miR-29-3p, and miR-34a were found to be highly expressed in the left ventricle of rats with CHF after MI. These miRNAs were preferentially encapsulated in exosomes derived from TNF-α-stressed cardiomyocytes and fibroblasts, which inhibit the translation of Nrf2 (Kelch-like ECH-associated protein 1-nuclear factor erythroid 2-related factor 2, an important transcription factor in the antioxidant defense mechanism) and the expression of antioxidant genes, thereby contributing to ischemic HF [96,97]. However, while these findings delineate a plausible EV-miRNA-mediated pathway impairing antioxidant defense in rodents, their direct translational relevance to human HF requires validation, as the complexity of oxidative stress regulation and the species-specificity of miRNA actions remain significant confounding factors.

### 5.4. EVs Mediate Heat-Shock-Protein Transport and Cardiac Hypertrophy

Diabetic cardiomyopathy (DCM), one of the leading causes of death in diabetics, is characterized by cardiac hypertrophy, necrosis, and diastolic dysfunction [98]. As the disease progresses, left ventricular hypertrophy and necrosis increase, ultimately leading to decreased left ventricular systolic function as evidenced by reduced ejection fraction [99].

Heat-shock-proteins (HSPs) are a defense mechanism under pathological conditions and play a key role in cellular resistance to stress [100]. Reduced HSP expression in diabetic patients leads to organ damage [101]. Among the heat shock protein family, HSP20 is the only HSP that responds to both acute and chronic elevations in blood glucose. Xiaohong et al. found that exosomes released from the cardiomyocytes of DCM mice carry a low level of HSP20, which contributes to myocardial remodeling and repression of endothelial cell proliferation, migration, and angiogenesis. The authors found that HSP20 increases exosome production in cardiomyocytes through direct interaction with TSG101, and that exosomes containing a higher level of HSP20 prevent cardiac hypertrophy and fibrosis [102]. In addition, Datta et al. demonstrated that HSP90 increased the synthesis and secretion of exosomal IL-6, and that during hypertrophy, HSP90 is packaged with IL-6 in the exosomes for transfer to CFs, activating STAT-3 signaling pathway and excessive collagen synthesis, ultimately resulting in severely impaired cardiac function during cardiac hypertrophy [103]. Lastly, HSP60, tightly attached to the exosome membrane within exosomes [104], has been detected in the plasma of rats with HF [105]. Se-Chan Kim et al. treated healthy rat cardiomyocytes with HSP60 purified from injured rat cardiomyocytes and confirmed that extracellular HSP60 mediates apoptosis via toll-like receptor (TLR)-4, suggesting that HSP60-mediated activation of TLR4 may be a mechanism for cardiomyocytes loss in HF [106]. Thus, exosomes associated HSP members seem to have opposite effects with HSP20 attenuating diabetic cardiac remodeling, HSP90 augmenting inflammation, and HSP60 mediating HF-related apoptosis. While these studies suggest distinct roles for EV-associated HSPs in diabetic cardiomyopathy, their clinical translation is hindered by the reliance on animal models, the lack of a unified mechanistic network explaining their differential sorting and functions, and the challenge of isolating specific HSP-carrying EV subpopulations for therapeutic targeting.

### 5.5. EVs Regulate Sympathetic Activity and HF

In recent years, researchers have demonstrated that neuroinflammation with evoked sympathoexcitation is critical in the development of CHF [107]. Early stages of sympathoexcitation are compensated for, but prolonged and sustained sympathoexcitation can lead to serious adverse cardiovascular events such as HF [108]. Mutually, the decline in cardiac function triggers the activation of the neurohormonal system and the development of cardiac hypertrophy [109]. Catecholamines are chronically over-secreted during the process, it binds to epinephrine receptors on cardiomyocytes and subsequently activates adenylate cyclase to increase cAMP-dependent protein kinase (PKA) signaling. Sustained cAMP-PKA signaling triggers maladaptive remodeling, including pathological cardiac hypertrophy, cardiomyocyte death, and cardiac fibrosis, and induces HF [110,111,112]. The rostral ventral lateral medulla (RVLM), located in the medulla oblongata, plays a crucial role in the regulation of the cardiovascular system and is closely related to the onset and progression of HF [113]. Since inflammation in the RVLM, a key region of sympathetic control, stimulates neuronal activity and increases sympathetic outflow. Interestingly, although peripheral inflammatory stimuli enhance the inflammatory response in RVLM, peripheral inflammatory factors do not directly affect RVLM due to the presence of the blood–brain barrier (BBB) [114]. In this context, EVs play a crucial role as a bridge between the central and peripheral systems [115]. Xiao et al. analyzed the differences between circulating exosomes and RVLM in CHF rats with miRNA microarray assays. The authors found that the expression of miR-214-3p was significantly upregulated, and in vitro cellular studies indicate that miR-214-3p enhances neuroinflammation, while let-7g-5p and let-7i-5p attenuate neuroinflammation. These results suggest that circulating exosomes are involved in enhancing the inflammatory response of RVLM through their cargo miRNAs, thereby augmenting sympathetic hyperactivity to contribute to CHF progression [116]. Overall, patients with HF influence the central inflammatory response by modulating circulating exosomes and their cargo, and monitoring changes in circulating exosomes also provides ideas for identifying targeted therapeutic strategies for HF.

### 5.6. EVs Mediate Reciprocal Regulation Between Cardiomyocytes and Fibroblasts in Cardiac Hypertrophy and HF

Studies have shown that downregulating the expression of phosphatase and tensin homolog (PTEN) or inhibiting the transforming growth factor-β (TGF-β)/small mother against decapentaplegic (SMAD) signaling pathway can reduce myocardial tissue necrosis and improve cardiac function [117,118]. miR-378 is abundantly expressed in the mammalian heart and acts as an inhibitor of the MAPK pathway and hypertrophy in cardiomyocytes [119]. Yuan J. et al. demonstrated that cardiomyocyte-derived exosomes can transport miR-378 into fibroblasts, targeting MKK6 and attenuating p38 MAPK-SMAD2/3 pathways, suppressing myocardial fibrosis at the early stage of cardiac hypertrophy induced by mechanical stress [120].

Interestingly, CFs can release exosomes that transport fibroblast miRNAs into cardiomyocytes to promote cardiomyocyte hypertrophy [7]. Bang C. et al. found that CFs secrete miRNA-rich exosomes that carry a relatively high abundance of miRNA passenger strands, or “star” miRNAs, which would normally be degraded intracellularly [121]. Utilizing confocal imaging and coculture assays, the authors identified fibroblast exosome-derived miR-21-3p (miR-21*) as a potent paracrine miRNA that induces cardiomyocyte hypertrophy. Specifically, when miR-21* is transported into cardiomyocytes, it targets SORBS2 (sarcoplasmic protein adsorption and SH3 domain-containing protein 2) and PDLIM5 (PDZ and LIM domain 5), promoting cardiac hypertrophy in the mouse pressure overload model. In addition, in Ang II infusion model, pharmacological inhibition of miR-21* attenuated cardiac hypertrophy. These findings suggest that CFs can secrete miRNA-rich exosomes, including miR-21* containing exosomes to mediate cardiomyocyte hypertrophy, and that fibroblast exosomal miR-21* is a possible biomarker and potential therapeutic target for pathological cardiac hypertrophy [121]. Although this signaling axis was validated in mouse models of pressure overload and Ang II infusion, the translational value in humans is uncertain. The expression level of miR-21* in circulating exosomes of HF patients and its correlation with disease severity need to be further verified, and the potential off-target effects of miR-21* inhibition should be evaluated in preclinical safety studies.

Tian C. et al. identified a pathogenic signaling axis in HF where CFs, stimulated by Ang II, release exosomes enriched with miRNA-27a guest strand (miR-27a*). Upon delivery to cardiomyocytes, this exosomal miR-27a* downregulates the target protein PDZ and LIM domain protein 5 (PDLIM5), triggering hypertrophic gene expression and contractile dysfunction. Inhibition of miR-27a* in vivo restored PDLIM5 levels, attenuated hypertrophy, and improved cardiac function. These findings suggest that miR-27a*-rich exosomes secreted by CFs may act as paracrine signaling mediators of cardiac hypertrophy and could potentially be new therapeutic targets [97].

Linmao Lyu et al. treated CFs with Ang II and found an increased release of exosomes from CFs, which in turn upregulated the expression of renin, angiotensinogen, AT1R and AT2R and enhance Ang II expression in cultured cardiomyocytes and exacerbated Ang II-induced pathological cardiac hypertrophy [122]. The further found increased levels of miR-217 expression in the hearts of CHF patients, in the exosomes of transverse aortic constriction (TAC) mouse cardiomyocytes, but not in the plasma of CHF patients. Overexpressed miR-217 was exported in exosomes from cardiomyocytes and transported to adjacent cells causing cardiac hypertrophy, fibrosis, and cardiac dysfunction. Furthermore, miR-217 appears to directly target the phosphatase and PTEN to mediate cardiac hypertrophy and cardiac dysfunction [123].

Peli1 is an E3 ubiquitin ligase involved in TGF-β1 activation of CFs [124]. A recent study showed that Peli1-induced cardiomyocyte exosomes are rich in miR-494-3p, which activates CFs by inhibiting PTEN and promoting the phosphorylation of Akt, SMAD2/3, and ERK, promoting the cardiac necrosis process and leading to HF. And inhibition of miR-494-3p from myocardium reduces pressure overload-induced cardiac fibrosis. This study provides a new idea for the treatment of HF [125].

### 5.7. EVs Mediate Adipose Regulation of Cardiac Hypertrophy and HF

Peroxisome Proliferator-Activated Receptor γ (PPARγ) is a master regulator of adipose tissue signaling with an essential role in insulin sensitivity, thus an important therapeutic target. However, when selective PPARγ agonist rosiglitazone (RSG) was used for treating diabetes, it had adverse cardiovascular effects, such as stimulating cardiac hypertrophy and oxidative stress that are independent of cardiomyocyte PPARγ. In an elegant study, Fang et al. used adipocyte and cardiomyocyte cocultures and found that stimulation of PPARγ signaling in adipocytes increased miR-200a expression and secretion. Delivery of miR-200a by adipocyte-derived exosomes to cardiomyocytes resulted in decreased TSC1 and subsequent mTOR activation, leading to cardiomyocyte hypertrophy. Treatment with an antagomir to miR-200a blunted this hypertrophic response in cardiomyocytes. In vivo, specific ablation of PPARγ in adipocytes was sufficient to blunt hypertrophy induced by RSG treatment [126]. Thus, adipose exosomes mediate the crosstalk between adipocytes and cardiomyocytes, contributing to cardiac remodeling.

In summary, EVs and their cargo miRNAs are extensively involved in the regulation of cardiac hypertrophy and HF under different pathological conditions (Table 2). Existing reports have focused largely on the roles of exosomal miRNAs of different cellular origin and their specific mechanisms, though common themes of EVs regulatory networks are yet to emerge. Although these preclinical studies elegantly demonstrate a causative role for specific EV cargos (e.g., miR-21*, miR-155) in hypertrophy, translating these findings to therapy faces hurdles. The pleiotropic effects of miRNAs raise concerns about off-target toxicity. Efficient and cardiac-specific delivery of inhibitory oligonucleotides (antagomirs) remains a technical challenge. Moreover, most models use acute pressure overload, which may not fully recapitulate the chronic, multifactorial nature of human HF.

## 6. EV-Based Therapeutic Approaches for Cardiac Hypertrophy and HF

### 6.1. EVs from Stem Cells

EVs derived from various populations of stem cells (e.g., iPSCs) have demonstrated benefits in the reversal of cardiovascular disease. For example, miR-21-rich exosomes from mouse cardiac fibroblast-derived iPSCs have been shown to protect against oxidative stress and cardiac remodeling in the peri-infarct region after AMI [127]. Oxidative stress upregulates miR-17 and miR-210 levels in CPC-derived exosomes, which can inhibit TGF-β-induced fibrosis [128]. In another study, Li Qiao et al. isolated exosomes from transplant-derived cardiac stromal cells from HF patients (F-EXO) or normal donor hearts (N-EXO). They found that the pathological condition of HF impairs the rejuvenation of cardiac-derived exosomal cargoes such as miRNAs, and that exosome-derived miR-21-5p contributes to exosome-mediated cardiac repair by enhancing cardiomyocyte survival via the PTEN/Akt pathway [129]. Additionally, Adamiak et al. compared the ability of mouse iPSC-derived EVs and their source iPSCs and found iPSC-derived EVs are safer and more effective for impart cytoprotective properties to cardiac cells in vitro and induce superior cardiac repair in vivo with regard to left ventricular function, vascularization, and amelioration of apoptosis and hypertrophy [130]. 

Doxorubicin (DOX) is an anthracycline anticancer drug widely used in clinical cancer chemotherapy, but its cardiotoxicity has become an increasingly serious problem. Long-term use of DOX can lead to myocardial disease, left ventricular dysfunction, and ultimately HF. In recent years, exosomes have gradually come into the public eye as diagnostic biomarkers and therapeutic means to reduce DOX-induced myocardial damage [131,132]. In 2020, Jie et al. demonstrated that stem cell-derived exosomes (TSC-Exos) improved cardiac function and alleviated DOX-induced cardiac injury through anti-inflammatory and anti-apoptotic effects [133]. Specifically, TSC-Exos blocked the DOX-activated NF-κB inflammatory signaling pathway, thereby inhibiting the inflammatory response. In addition, TSC-Exos downregulated the expression of the pro-apoptotic mediator miR-200b and increased the expression of the anti-apoptotic mediator E-cadherin transcriptional repressor Zeb1, thereby reducing cardiomyocyte apoptosis. Junfeng et al. also demonstrated that TSC-Exos alleviated DOX-induced cardiac injury through anti-apoptotic effects and improved mitochondrial fusion by increasing Mfn2 expression, providing a potential therapeutic option for the treatment of HF [134].

In addition, embryonic stem cell (ESC)-derived exosomes are highly enriched with FGF2 protein, which can promote myocardial angiogenesis and alleviate transverse aortic constriction (TAC)-induced HF, which is also a new option for the treatment of HF [135]. Furthermore, a recent study demonstrates that exosomes derived from induced pluripotent stem cells (iPSC-Exos) can be effectively loaded with Necrostatin-1 (Nec-1) to attenuate heart failure by targeting the PARP1/AIFM1 axis, thereby restoring mitochondrial function and reducing oxidative stress in cardiomyocytes [136].

### 6.2. EVs from CPCs and CDCs

It has been shown that exosomes derived from CPCs are involved in cardio-protection and repair [137,138]. Xiao J. et al. showed that oxidative stress increased exosome release from CPCs, and that these exosomes inhibited H_2_O_2_-induced apoptosis in H9C2, but that the above biological effects were more effective in these exosomes treated with H_2_O_2_. They further demonstrated that oxidative stress increased miR-21 levels in exosomes, which inhibited H_2_O_2_ apoptosis by targeting Programmed Cell Death 4 (PDCD4), and therefore concluded that CPC-derived exosomes counteracted cardiomyocyte apoptosis through both pathways [139]. It is known that inhibition of cardiomyocyte apoptosis attenuates compensatory hypertrophy in surviving cardiomyocytes [140]. Based on the above studies, it can be concluded that CPCs-derived exosomes inhibit the process of cardiac hypertrophy after MI.

Cardiosphere-derived cells (CDCs) have been shown to reduce cardiomyocyte death, promote tissue regeneration, stimulate angiogenesis, inhibit interstitial fibrosis, and suppress inflammation [127,128]. These therapeutic effects can be largely recapitulated by CDCs-derived exosomes (CDC-Exos) [129]. Various heart homing peptides (HHPs) have been engineered to display on the surface of exosomes, such as by expressing HHP fused in-frame with transmembrane proteins in exosome-producing cells [130]. Bittle et al. isolated CDC-Exos from humans and injected them into the myocardium of a young pig stress model. They found that CDC-Exo enhanced myocardial contractility and inhibited apoptosis, ultimately preventing right ventricular dysfunction [131]. Liang Mao et al. engineered the expression of HHPs displayed on the surface of exosomes derived from human CDCs, termed HHP-Exos. They demonstrated that intravenous administration of HHP-Exo significantly improved cardiac function in a mouse model of TAC-induced cardiac hypertrophy and HF, and that HHP-Exo reduced left ventricular hypertrophy more than control exosomes (CON-Exos) by inhibiting p-STAT3, p-ERK1/2, β-MHC, BNP, GP130, and p-AKT [132]. Meanwhile, miR-148a was selectively enriched in CDC-Exos and, after systemic delivery, significantly restored TAC-induced miR-148a downregulation and ameliorated cardiac hypertrophy in the TAC model. They concluded that HHP-Exo preferentially targets the heart and improves the therapeutic effect of CDC-Exos on cardiac hypertrophy, and that this therapeutic effect is likely due to miR-148a-mediated inhibition of GP130 and inhibition of the STAT3/ERK1/2/AKT signaling pathway, resulting in improved cardiac function and myocardial hypertrophy [132]. Feng Huang et al. injected YF1, a CDC-Exo-derived non-coding RNA with 56 nucleotides, into HCM mice and found that HCM mice injected with YF1 had improved ambulation, cardiac hypertrophy, and myocardial fibrosis. At the same time, they found that these mice had reduced peripheral neutrophils mobilization and pro-inflammatory monocytes, as well as reduced infiltration of macrophages into their hearts. Therefore, they conclude that exosome-derived YF1 from CDCs modulates pathways associated with immune disorders and inflammatory responses, reversing hypertrophic and fibrotic signaling pathways associated with HCM, which makes YF1 a potentially viable novel therapeutic candidate for the treatment of HCM [133]. The therapeutic efficacy of CDC-derived EVs (CDC-EVs) can be enhanced by engineering. Mao et al. engineered CDC-EVs to display a heart-homing peptide (HHP), resulting in a three-fold increase in cardiac accumulation and superior amelioration of hypertrophy in a pressure-overload model compared to unmodified EVs [141]. This exemplifies the promise of targeting strategies.

### 6.3. EVs from Endothelial Progenitor Cells

Adverse cardiac remodeling and HF is frequently accompanied by vascular dysfunction, which involves exosomes and their cargoes circRNAs [59,124,125]. Huihua et al. found that circ_0018553 is enriched in endothelial progenitor cell (EPC)-derived exosomes but downregulated in Ang II-induced hypertrophic cardiomyocytes. In addition, circ_0018553 acts as a sponge for miR-4731, which directly inhibits sirtuin 2 (SIRT2) expression. In a cellular model of Ang II-induced cardiac hypertrophy, overexpression of circ_0018553 attenuated whereas silencing circ_0018553 worsened cardiac hypertrophy in miR-4731 dependent manner. miR-4731 overexpression attenuated the protective effects of circ_0018553 overexpression, and SIRT2 silencing attenuated the protective effects of miR-4731 inhibition. Thus, EPC-derived exosomal circ_0018553 inhibits Ang II-induced cardiac hypertrophy via the miR-4731/SIRT2 pathway [126]. Khan et al. demonstrated that EPC-EV administration into an infarcted heart increases neovascularization, inhibits cardiomyocyte apoptosis, reduces scar size, and improves left ventricular function, and that IL-10 is critical for EPC-EVs’ beneficial effects by repressing aberrant activation of cardiac inflammation [142]. In addition to targeting ECs, EPC-EVs were also found miR-1246 and miR-1290 in EPC-derived exosomes to enhance vascularization by acting on fibroblasts to promote fibroblast-to-endothelial transition [143].

### 6.4. EVs from MSCs

MSCs are pluripotent stem cells derived from various tissues and organs such as adipose tissue, muscle, and bone marrow. MSCs protect the heart by inhibiting apoptosis, suppressing inflammation and promoting myocardial angiogenesis [144].

Researchers have found that MSCs can protect and repair the heart by secreting exosomes, therefore, MSC exosomes have been widely studied as an alternative to MSCs, and their cardioprotective and therapeutic effects are gradually being confirmed [145,146]. Yuto et al. found that the cardioprotective effect of intravenously injected mesenchymal stem cells in a mouse model of pressure-overload HF actually relies heavily on the circulating adipocyte-secreted factor, adiponectin. The injected MSCs function through exosomes. The unique glycosylphosphatidylinositol-anchored T-adhesin binding on MSCs interacts with plasma adiponectin in the heart and vascular endothelium to stimulate exosome biogenesis and secretion, thereby amplifying the therapeutic effects of MSC-exosomes [147,148]. Therefore, induction of lipocalin production is a promising therapeutic strategy [149].

It is well known that cardiomyocytes in patients with HF are usually hypertrophied, and the hypertrophied cells lead to an increase in myocardial energy demand, which in turn leads to an imbalance in myocardial blood supply, so the promotion of angiogenesis is an important node in the prevention and treatment of HF [150]. Recently, Zicheng et al. demonstrated that during CHF, exosomal miR-1246 released from human umbilical cord mesenchymal stem cells (hucMSCs) specifically targets serine protease 23 (PRSS23), blocking the activation of Snail signaling/α-smooth muscle actin signaling, which in turn inhibits the expression of endothelial cell marker CD31 and promotes angiogenesis and attenuate hypoxia-induced myocardial tissue injury. Overall, their findings suggest that the exosomal miR-1246 produced by hucMSCs protects the heart from failure by targeting PRSS23 [151].

### 6.5. Other Kinds of EVs

Severe myocardial necrosis after myocardial infarction (MI) is an important factor leading to HF. Therefore, researchers are committed to developing effective means to prevent myocardial necrosis after myocardial infarction in order to reduce the occurrence of HF after MI [152].

Ischemic preconditioning (IPC) protects patients with coronary artery disease with multiple lesions from myocardial injury due to ischemia-reperfusion injury (IRI) [153]. Recently, researchers transferred exosomes produced by normal rats undergoing IPC to post-infarction HF rats. They found that these exosomes activated pro-survival protein kinase, which exerted a protective effect against cardiac IRI through phosphorylation of ERK and AKT. This study provides new therapeutic ideas for patients with IRI complicated by CHF [154].

Exosomes carrying nucleic acids are known to be a promising means of targeted therapy. However, preferential uptake by the liver and spleen limits the clinical application of exosomes. Recently, investigators pre-injected exosomes loaded with small interfering RNA (siRNA) targeting lectin heavy chain (Cltc) prior to the injection of therapeutic exosomes in order to block the endocytosis of therapeutic exosomes by macrophages. This approach allows for more accurate delivery of therapeutic exosomes to target organs such as the heart, significantly improving exosome-associated gene therapy regimens [155].

Microneedle (MN) patches deliver bioactive substances in a painless transdermal manner, which can achieve therapeutic effects on diseases [156]. Researchers combined EVs containing miR-29b from human umbilical cord mesenchymal stem cells (HucMSCs) with MNs and implanted them into the myocardial infarction area. They found that these modified MNs could alleviate cardiac necrosis by inhibiting the TGF-β/SMAD3 signaling pathway in CFs [157].

Chronic inflammation plays a crucial role in the development of HF [158]. To date, several studies have shown that patients with HF have an increased number and hyperactivity of peripheral blood mononuclear cells (PBMCs), which promotes the release of inflammatory factors and tumor necrosis factor alpha (TNF-α), thereby accelerating the progression of HF [159]. In a recent study, EVs from healthy donor plasma were shown to regulate paracrine secretion of PBMCs in patients with CHF, decrease miRNA-126 expression, and reduce the release of inflammatory factors, resulting in cardioprotective effects [79].

### 6.6. Engineering Strategies and Clinical Translation of EV-Based Therapeutics

The therapeutic potential of native EVs is often limited by non-specific biodistribution, rapid clearance, and insufficient cargo loading. To overcome these hurdles, a variety of engineering strategies have been developed to enhance the targeting, efficacy, and manufacturability of EV-based therapeutics for cardiac diseases [160,161].

To improve the therapeutic efficacy and targeting of EVs, various engineering modification strategies have been developed in recent years, including surface modification and cargo loading. For surface modification, expressing heart-homing peptides (e.g., HHP, CP05) on the EV membrane can significantly enhance the enrichment efficiency of EVs in myocardial tissue [162,163]. Mao L et al. intravenously injected HHP-modified CDC-derived EVs into TAC mice, and the myocardial targeting efficiency was increased three-fold, with a significantly better improvement effect on cardiac hypertrophy than unmodified EVs [141]. Tong L. et al. developed a targeted engineered extracellular vesicle platform, miR30d-mEVsIMTP, by dual-engineering milk-derived EVs with an ischemic myocardium-targeting peptide (IMTP) and encapsulating therapeutic miR-30d. This system achieved efficient cardiac delivery, attenuated pathological hypertrophy, and restored cardiac function in murine models. Mechanistically, it identified GRK5 as a novel target of miR-30d, through which it counteracts maladaptive remodeling. This work establishes a translatable paradigm of synergistic EV engineering for precision therapy of cardiovascular disease [164]. In addition, a study developed a cardiac-targeted liposomal system (Lipo@miR-185-5p inhibitor) encapsulating an exosome-derived miRNA-185-5p inhibitor for treating dilated cardiomyopathy. The strategy effectively delivered its cargo to the heart, significantly reducing both apoptosis and cuproptosis in cardiomyocytes in vitro. In a doxorubicin-induced dilated cardiomyopathy mouse model, this targeted therapy improved cardiac function, evidenced by a 27.3% increase in LVEF, and reduced myocardial fibrosis by 36.5%, highlighting its potential as a novel targeted molecular therapy for heart failure [165]. For instance, exosomes displaying a platelet-derived growth factor (PDGF) receptor-recognizing peptide showed improved retention in infarcted myocardium [166]. While these preclinically validated, engineered EV platforms demonstrate significant potential for targeted cardiac therapy, their ultimate clinical translation will depend on resolving critical challenges related to standardized manufacturing, consistent in vivo targeting efficiency, and comprehensive long-term safety and efficacy assessment.

Active loading methods such as electroporation, sonication, or transfection of donor cells are used to enrich EVs with therapeutic miRNAs (e.g., miR-21, miR-148a), siRNA, or small molecules [166]. Engineered exosomes overexpressing cytoprotective proteins like HSP20 have shown enhanced efficacy in inhibiting hypertrophy [102]. Incorporating EVs into hydrogels or microneedle patches allows for sustained local release at the cardiac site, prolonging therapeutic effects and reducing systemic exposure [157]. A recent study demonstrated an injectable, conductive, and oxygen-generating biomaterial scaffold by integrating catalase, gold nanoparticles, and MSC-derived exosomes into an Alg-Fib hydrogel; in a rat MI model, this multifunctional hydrogel system enhanced angiogenesis, reduced apoptosis, and improved cardiac function by concurrently delivering oxygen, providing electrical cues, and releasing reparative exosomal cargo [167].

Transitioning from laboratory-scale production to good manufacturing practice (GMP)-compliant, large-scale EV manufacturing remains a major bottleneck. Standardization of potency assays (e.g., based on miRNA content or surface markers) is critical for quality control. Key translational challenges include defining the mode of action, ensuring batch-to-batch consistency, assessing long-term safety (especially with engineered EVs), and navigating evolving regulatory pathways for EV-based products as drug delivery vehicles or biological therapeutics [168,169]. The convergence of EV biology with bioengineering and regulatory science will be pivotal in realizing the clinical promise of EV-based therapies for cardiac hypertrophy and HF.

## 7. Conclusions and Prospect

In conclusion, the review has focused on delineating the dual roles of EVs as both mediators of pathological remodeling and promising therapeutic vectors in the context of cardiac hypertrophy and HF. We have highlighted the cell-type-specific EV trafficking that underlies key processes like inflammation, fibrosis, and neurohormonal activation. The rapid evolution of EV engineering, encompassing targeting, cargo loading, and hybrid delivery systems, is poised to transform these natural nanocarriers into precision medicines for the heart. The accumulating positive outcomes from preclinical and early-stage clinical studies of EV-based therapies, as summarized in Table 3, underscore this transformative potential.

However, the EV research also has challenges. EVs are heterogeneous, they come from a variety of sources, almost all cell types release EVs, and their composition varies depending on the parent cell and its physiological state. The size and content variability of EVs have a large size range and carry a wide variety of cargoes (proteins, lipids, nucleic acids), making them difficult to standardize. Currently, there is a lack of standardized methods for the separation and purification of EVs, and the co-isolation of non-EV components (e.g., lipoproteins, protein aggregates) may make results inaccurate. At the same time, identification of specific EV markers for different isoforms is complex due to overlapping surface proteins. Additionally, because EVs have complex cargo and multifunctional effects, it can be difficult to decipher the exact mechanism by which EVs affect recipient cells. Further, mass production of EVs for therapeutic use has not yet been overcome both technically and economically, its safety and efficacy are uncertain, and extensive preclinical and clinical validation is required to ensure the safety, stability, and targeted delivery of EV-based therapies.

Future research should leverage multi-omics to decode patient-specific EV signatures, develop next-generation smart EVs, and explore synergistic combination therapies. By bridging deep mechanistic insights with innovative bioengineering, EV-based diagnostics and therapeutics hold immense potential to revolutionize the management of cardiac hypertrophy and heart failure, ultimately fulfilling their promise as precision nanomedicines for the heart.

## Figures and Tables

**Figure 1 ijms-27-00095-f001:**
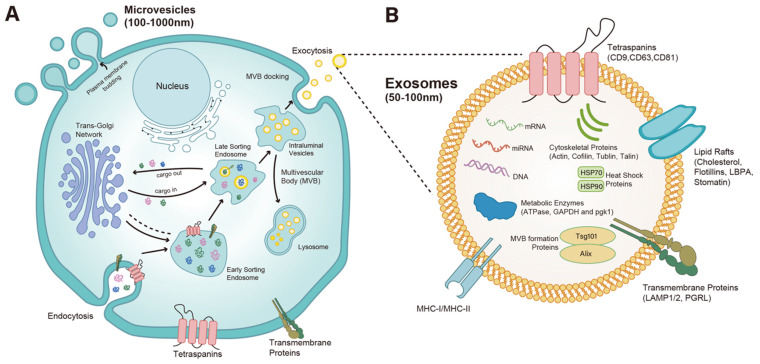
Biogenesis and cargo components of exosomes. (**A**). Cell membrane invaginates inwardly to form endosomes, then ESEs encapsulate bioactive substances via material exchange with the trans-Golgi network (TGN). When ESEs mature into LSEs, the sorting cargoes on the endosome membrane drive inward invagination of the membrane into endosomal lumen to form ILVs. Finally, endosomes containing ILVs (i.e., MVBs) fuse with the cellular membrane and release ILVs into extracellular space as exosomes. (**B**). Exosomes have a lipid bilayer structure containing a range of transmembrane proteins and receptors, including tetraspanins (e.g., CD9, CD63, and CD81), transmembrane proteins (e.g., LAMP1/2 and PGRL), lipid rafts (e.g., Cholesterol, Flotillins, LBPA, and Stomatin) and immunomodulatory molecules (e.g., MHC-I and II). Within the exosome lumen are molecules that carry endosomal pathways indicative of their origin, such as cytoskeletal proteins (e.g., Actin, Cofilin, Tublin, and Talin), heat shock proteins (e.g., HSP70 and HSP90), metabolic enzymes (e.g., ATPase, GAPDH, and pgk1), proteins involved in MVB formation (e.g., TSG101 and Alix) and RNA molecules (miRNAs, mRNAs, and other short RNAs). Exosomes can modulate the bioactivity of other cells by binding to their surface receptors, fusing with the cell membrane, or entering the cell via endocytosis.

**Figure 2 ijms-27-00095-f002:**
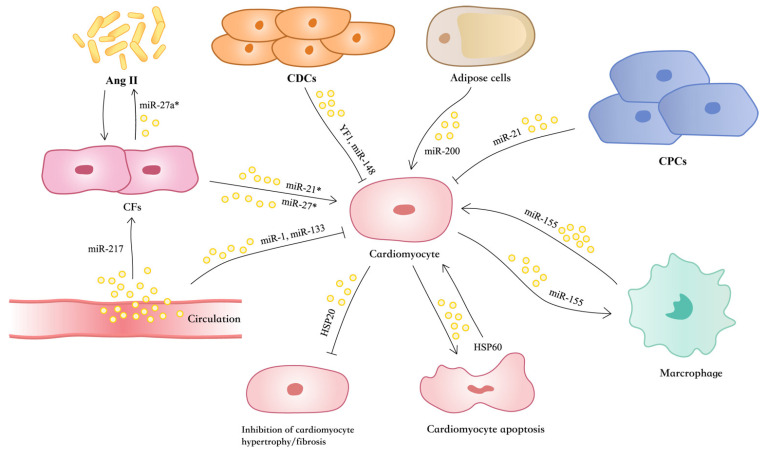
Exosomes function in cardiac hypertrophy. Circulating exosomal miR-1 and miR-133 inhibits cardiac hypertrophy. Exosomal miR-155 produced by AngII-induced hypertrophic cardiomyocytes is involved in the inflammatory response of macrophages. Exosomal miR-21* secreted by CFs induce cardiomyocyte hypertrophy. CFs secreted miR-27a*-rich exosomes in response to Ang II stimulation, which leads to myocardial hypertrophy by inhibiting PDLIM5 and leads to increased Ang II production thereby exacerbating Ang II-induced pathological cardiac hypertrophy. Circulating exosomal miR-217 enhanced cardiac fibroblast multiplication. CDCs’ exosomal miR-148a improves cardiac function and reduces left ventricular hypertrophy in TAC mice. CDCs’ exosomal YF1 inhibits cardiomyocyte hypertrophy. Exosomal miR-378 from cardiomyocytes reduces myocardial fibrosis. Macrophage exosomal miR-155 is transduced into the myocardium of uremic mice and acted on FOXO3a to promote cardiomyocyte hypertrophy and fibrosis in UCM. CPCs’ exosomal miR-21 inhibits cardiac hypertrophy after MI. Adipocyte exosomal miR-200 inhibits TSC1 and activates mTOR, leading to cardiomyocyte hypertrophy. HSP20 interacts with TSG101 to promote exosome production in cardiomyocytes, and exosomes containing higher levels of HSP20 prevent cardiac hypertrophy and myocardial fibrosis. Exosomal HSP60 from injured rat cardiomyocytes mediates apoptosis via TLR-4.

**Table 1 ijms-27-00095-t001:** Clinical and translational studies of EV-related biomarkers and therapies in pathological cardiac hypertrophy and HF.

Etiology/HF Context	Study Population/Design	EV Source	EV Cargo/Marker(s)	Key Findings/Clinical Relevance	Ref.
Chronic HF (mixed ischemic/non-ischemic)	HF patients vs. healthy controls, cross-sectional	Plasma exosomes	miR-146a, miR-486, miR-16 (ratios miR-146a/miR-16, miR-486/miR-16)	Increased exosomal miRNA ratios in HF vs. controls added diagnostic value	[58]
Acute HFrEF	Hospitalized acute HFrEF patients vs. healthy volunteers	Serum exosomes	miR-92b-5p, miR-192-5p, miR-320a	Selective elevation of exosomal miR-92b-5p in acute HFrEF	[59]
Acute HF and cardiac fibrosis	Patients with acute HF vs. controls	Plasma exosomes	miR-21, miR-425, miR-744	Imbalanced exosomal miR-21/miR-425/miR-744 associated with fibrosis and HF progression	[62]
Post-AMI risk of ischemic HF	Post-AMI patients with vs. without HF during follow-up	Serum exosomes	miR-192, miR-194, miR-34a (p53-responsive miRNAs)	p53-related exosomal miRNAs predicting post-AMI ischemic HF	[63]
Chronic HF severity/prognosis	HF cohort with longitudinal outcome follow-up	Plasma endothelial EVs and apoptotic microparticles	CD31+/CD144+ EVs, CD144+/CD31+/Annexin V+ microparticles	Endothelial EVs/microparticles associated with HF severity and adverse prognosis	[64,65,66]
PPCM-HF	PPCM patients vs. DCM patients; serial sampling	Peripheral blood exosomes	miR-146a	Exosomal miR-146a as PPCM-specific diagnostic and prognostic marker	[67]
Maternal obesity—offspring cardiac programming	Umbilical cord samples from obese mothers; in vitro cardiomyocyte assays	Umbilical cord serum EVs	miR-142, miR-17 (and other cargo)	Cord EVs from obese mothers inducing pro-hypertrophic responses in neonatal cardiomyocytes	[69]
MMVD and chronic HF (veterinary)	Dogs with MMVD, MMVD-CHF and aging controls	Plasma exosomes vs. total plasma	miR-9, miR-495, miR-599, miR-181c and others	Exosomal miR-181c/miR-495 as MMVD-CHF–related biomarkers (veterinary)	[70]
Diabetic cardiomyopathy in T2DM	T2DM patients with vs. without DCM	Plasma (circulating miRNAs)	miR-21	Reduced plasma miR-21 indicating DCM in T2DM	[74]
HCM (patient-derived cellular model)	ACTC1-mutant HCM hiPSC-CMs vs. control hiPSC-CMs	hiPSC-cardiomyocyte-derived EVs	A total of 12 snoRNAs (10 SNORDs, 2 SNORAs) and enriched transcripts	Distinct snoRNA/transcript signature in HCM cardiomyocyte EVs as biomarker candidates	[75]
Chronic HF—EV-based immunomodulation	PBMCs from chronic HF patients treated in vitro	EVs from healthy donor plasma	Global EV cargo; miR-126 changes in PBMCs	Healthy donor EVs modulating CHF PBMC inflammation; immunomodulatory therapy proof-of-concept	[79]

**Table 2 ijms-27-00095-t002:** EVs and cargoes function in cardiac hypertrophy and HF through different pathways.

EVs Origin	Cargoes	Target Gene/Pathways	Outcomes/Functions	Ref.
SHR serum	—	Carry angiotensinogen, renin, and ACE into cardiomyocytes to increase autocrine secretion of Ang II	Cardiac hypertrophy ↑	[88]
Indoxyl sulfate-induced cardiac microvascular endothelial cell (CMEC)	miR-27a-5p	Target USF2/FUT8 axis in cardiomyocytes	Reactive oxygen species (ROS) ↑ Cardiac apoptosis ↑ Cardiac hypertrophy ↑	[92]
Ang II-induced hypertrophic cardiomyocytes	miR-155	Induce phosphorylation of ERK, JNK and p38	Macrophages inflammatory response ↑	[94]
MI mouses cardiomyocytes	lncRNA H19	NF-κB signaling pathway	Cardiac apoptosis inflammation ↓ Cardiac abnormal remodeling ↓ Exaggerated hypertrophy of ischemia-reperfused myocardium due to H19 knockout	[95]
TNF-α-stressed cardiomyocytes and fibroblasts	miR-27a, miR-29-3p, and miR-34a	Inhibit the translation of Nrf2 and the expression of antioxidant genes	Ischemic heart failure ↑	[96,97]
CFs/CHF rat	miR-27a guest strand	Inhibit PDLIM5 translation in cardiomyocytes; Induce AngII expression	Cardiac hypertrophy ↑ Myocardial contractility ↓	[97]
Cardiomyocytes/Mice	HSP20	Interact with TSG101	Exosomes biogenesis and release ↑ Cardiac hypertrophy ↓ Myocardial apoptosis ↓ Myocardial fibrosis ↓	[102]
Cardiomyocytes	HSP90	Activate STAT-3 signaling pathway	Cardiac hypertrophy ↑ Cardiac function ↓	[103]
Serum/CHF rats	miR-214-3p	Enhance the inflammatory response of RVLM, hyperactive nervous system	Heart failure ↑	[116]
CFs/TAC mice	miR-21-3p	Target SORBS2 and PDLIM5	Cardiac hypertrophy ↑	[121]
Angiotensin II treated mice CFs	——	AngII enhances exosomes release via targeting AT1R and AT2R; exosomes upregulate RAS via activation of MAPK and Akt	Cardiac hypertrophy ↑	[122]
CHF patients/TAC mice	miR-217	Target PTEN	Heart size ↑ Cardiac hypertrophy ↑ Myocardial fibrosis ↑	[123]
Peli1-induced cardiomyocyte	miR-494-3p	Inhibit PTEN and promote Akt, SMAD2/3, and ERK signaling pathway	Cardiac necrosis ↑ Heart failure ↑	[125]
Adipocytes	miR-200a	Decrease TSC1 and subsequent mTOR activation in cardiomyocytes	Cardiac hypertrophy ↑	[126]

**Table 3 ijms-27-00095-t003:** Therapeutical potentials of EVs and cargoes in cardiac hypertrophy and HF.

EVs Origin	Cargoes	Target Gene/Pathways	Outcomes/Functions	Ref.
Cardiomyocytes in mammals	miR-378	Target MKK6, attenuate p38 MAPK phosphorylation; downregulate p38 MAPK-Smad2/3 pathway	Cardiac hypertrophy ↓ Myocardial fibrosis ↓	[120]
Transplant-derived cardiac stromal cells from HF patients	miR-21-5p	Inhibit PTEN, enhance Akt kinase activity	Cardiomyocyte survival ↑ Heart failure ↓	[129]
Stem cell-derived exosomes (TSC-Exos)	N/A	Anti-inflammation: Block DOX-activated NF-κB inflammatory signaling pathway Anti-apoptosis; downregulate miR-200b expression and increase Zeb1 expression	Inflammation ↓ Myocardial apoptosis ↓ DOX-induced cardiac injury ↓	[133]
TSC-Exos	N/A	Improve mitochondrial fusion and increase Mfn2 expression	DOX-induced cardiac injury ↓	[134]
CPCs	miR-21	Target PDCD4 Inhibit apoptosis of H9C2	Cardiac hypertrophy ↓ Myocardial apoptosis ↓ Myocardial fibrosis ↓ Heart failure ↓	[139]
EPC	Circ_0018553	Target miR-4731/SIRT2 signaling pathway	Cardiac hypertrophy ↓	[170]
CDCs/Human	miR-148a	Inhibit GP130 Suppress STAT3/ERK1/2/AKT pathway	Cardiac hypertrophy ↓	[141]
CDCs	YF1	Downregulate JNK and Smad2 phosphorylation; attenuate CXCL1 expression in cardiomyocytes	Cardiac hypertrophy ↓ Myocardial fibrosis ↓ Myocardial apoptosis ↓ Myocardial fibrosis ↓	[171]
hucMSCs	miR-1246	Inhibit Snail/alpha-smooth muscle actin signaling	Angiogenesis ↑ Heart failure ↓	[151]
Normal rats undergoing IPC	pro-survival protein kinase	Phosphorylation of downstream ERK and AKT	Restore cardioprotection in HF after MI	[154]
EVs from HucMSCs	miR-29b	Inhibit TGF-β/SMAD3 signaling pathway	Cardiac necrosis ↓	[157]
EVs from healthy donor plasma	N/A	Decrease miRNA-126 expression; reduce the release of inflammatory factors	Cardioprotective effects ↑	[79]

## Data Availability

No new data were created or analyzed in this study. Data sharing is not applicable to this article.

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
