# Peer review of "Potential Biomarker and Therapeutic Tools for Pathological Cardiac Hypertrophy and Heart Failure: Extracellular Vesicles"

_ijms, 2025, doi:10.3390/ijms27010095_

Round 1
Reviewer 1 Report
Comments and Suggestions for Authors
The authors of the manuscript entitled: “Potential Biomarker and Therapeutic Tools for Pathological Cardiac Hypertrophy and Heart Failure: Extracellular Vesicles” presented interesting information related to the role of extracellular vesicles (EVs) in cardiac hypertrophy and heart failure (HF). Furthermore, it presents therapeutic, prophylactic, and diagnostic use of EVs I these pathological conditions. The manuscript is comprehensive, informative, well-structured ad accompanied with appropriate figures and tables. However, several minor issues need to be resolved.
- Abstract would benefit from a clearer statement highlighting the novelty of this review paper.
- In the Introduction section, improve transition and flow between the general background on the EVs and HF and the specific aim of this manuscript would improve readability.
- When discussing HF classification please provide full HF classification, which includes all its categories based on the ejection fraction of the left ventricle, according to the latest updated international guidelines.
- Is there any evidence related to other causes of HF and EVs, apart from those you already discussed? If those studies exist, please add them I appropriate section.
- In sections reviewing findings from clinical and preclinical studies, adopt a more critical approach when discussing the findings rather than a purely narrative one.
- Please create a dedicated table summarizing the key findings from clinical studies discussed in the manuscript.
- Technical suggestions: All abbreviations need to be defined when first mentioned in the main text, such as EV and HF. Please revise the citation formatting throughout the manuscript for technical accuracy and consistency. In particular, the citation at line 267 appears incorrectly formatted and should be corrected according to the journal’s reference style. Uniform the font size in Table 1, especially in column references.
Author Response
Response Letter
Potential Biomarker and Therapeutic Tools for Pathological Cardiac Hypertrophy and Heart Failure: Extracellular Vesicles
Manuscript ID: ijms-4023259
Dear Reviewer.
On behalf of my co-authors, we thank you very much for giving us an opportunity to revise our manuscript, we appreciate the editor and reviewers very much for their insightful and constructive comments, which have been invaluable in guiding a comprehensive revision of our manuscript entitled “Potential Biomarker and Therapeutic Tools for Pathological Cardiac Hypertrophy and Heart Failure: Extracellular Vesicles (Manuscript Number: ijms-4023259).” We have carefully considered all comments and made the necessary revisions on the cover letter and the manuscript. The following provides detailed responses to each of the comments, with the reviewer’s comments presented in black font, our response in blue font, and all revisions in the manuscript are highlighted in red. Please see our responses in the attachment.
We believe these extensive revisions significantly enhancing the manuscript's originality, analytical rigor, and value to the field. We would like to express our great appreciation to you for your comments on our paper. Looking forward to hearing from you.
Thank you and best regards.
Sincerely yours.
Min Cheng, MD, Ph.D.
Department of Cardiology, Union Hospital
Tongji Medical College, Huazhong University of Science and Technology,
Wuhan 430022, P R China.
E-mail: min_cheng@hust.edu.cn

Reviewer 2 Report
Comments and Suggestions for Authors
Although the manuscript addresses a scientifically relevant and timely topic, namely the role of extracellular vesicles (EVs) in the pathogenesis and treatment of pathological cardiac hypertrophy and heart failure, including the regulatory mechanisms of EVs in cardiomyocytes and their potential diagnostic and therapeutic applications, I must emphasize that the field has already been extensively reviewed in recent years. Multiple comprehensive review articles published between 2020 and 2025 cover the same thematic scope. Representative examples include:
- Fu S, Zhang Y, Li Y, Luo L, Zhao Y, Yao Y. Extracellular vesicles in cardiovascular diseases. Cell Death Discov. 2020, 6, 68. https://doi.org/10.1038/s41420-020-00305-y
- Chen M, Wu Y, Chen C. Extracellular Vesicles as Emerging Regulators in Ischemic and Hypertrophic Cardiovascular Diseases: A Review of Pathogenesis and Therapeutics. Med. Sci. Monit. 2025, 31, e948948. https://doi.org/10.12659/MSM.948948
- Hu H, Wang X, Yu H, Wang Z. Extracellular vesicular microRNAs and cardiac hypertrophy. Front. Endocrinol. (Lausanne) 2025, 15, 1444940. https://doi.org/10.3389/fendo.2024.1444940
- Li H, Li Z, Fu Q, Fu S, Xiang T. Exploring the landscape of exosomes in heart failure: a bibliometric analysis. Int. J. Surg. 2025, 111(5), 3356–3372. https://doi.org/10.1097/JS9.0000000000002248
- Guerricchio L, Barile L, Bollini S. Evolving Strategies for Extracellular Vesicles as Future Cardiac Therapeutics: From Macro- to Nano-Applications. Int. J. Mol. Sci. 2024, 25(11), 6187. https://doi.org/10.3390/ijms25116187
This review lacks a clear added value compared with earlier publications addressing the topic of EVs in cardiology. The manuscript has a predominantly summary-like character and largely reiterates findings already presented in previous works. Additionally, nearly 50% of the cited literature dates from before 2016–2017, which significantly reduces the current relevance of the review. The article therefore requires substantial revision. In particular, it would be advisable to expand Section 6, which discusses EV-based therapeutic approaches for cardiac hypertrophy and heart failure. As indicated by the cited literature, this area encompasses the most recent research findings and could provide genuine added value to the review, especially if supplemented with a critical analysis of current trends, limitations, and translational perspectives. Nevertheless, the authors should carefully consider the aforementioned review articles to avoid redundancy and to focus instead on aspects that have not yet been thoroughly analyzed. Therefore, from the perspective of originality and contribution to the field, the manuscript does not fulfill the journal’s criterion of novelty.
Author Response
Response Letter
Potential Biomarker and Therapeutic Tools for Pathological Cardiac Hypertrophy and Heart Failure: Extracellular Vesicles
Manuscript ID: ijms-4023259
Dear Reviewer.
Thank you very much for your insightful and constructive comments, which have been invaluable in guiding a comprehensive revision of our manuscript entitled “Potential Biomarker and Therapeutic Tools for Pathological Cardiac Hypertrophy and Heart Failure: Extracellular Vesicles (Manuscript Number: ijms-4023259).” We have carefully considered all comments and made the necessary revisions on the cover letter and the manuscript. The following provides detailed responses to each of the comments, with the reviewer’s comments presented in black font, our response in blue font, and all revisions in the manuscript are highlighted in red.
Response Letter
Potential Biomarker and Therapeutic Tools for Pathological Cardiac Hypertrophy and Heart Failure: Extracellular Vesicles
Manuscript ID: ijms-4023259
Dear Reviewer.
On behalf of my co-authors, we thank you very much for giving us an opportunity to revise our manuscript, we appreciate the editor and reviewers very much for their insightful and constructive comments, which have been invaluable in guiding a comprehensive revision of our manuscript entitled “Potential Biomarker and Therapeutic Tools for Pathological Cardiac Hypertrophy and Heart Failure: Extracellular Vesicles (Manuscript Number: ijms-4023259).” We have carefully considered all comments and made the necessary revisions on the cover letter and the manuscript. The following provides detailed responses to each of the comments, with the reviewer’s comments presented in black font, our response in blue font, and all revisions in the manuscript are highlighted in red.
Response Letter
Potential Biomarker and Therapeutic Tools for Pathological Cardiac Hypertrophy and Heart Failure: Extracellular Vesicles
Manuscript ID: ijms-4023259
Dear Reviewer.
On behalf of my co-authors, we thank you very much for giving us an opportunity to revise our manuscript, we appreciate the editor and reviewers very much for their insightful and constructive comments, which have been invaluable in guiding a comprehensive revision of our manuscript entitled “Potential Biomarker and Therapeutic Tools for Pathological Cardiac Hypertrophy and Heart Failure: Extracellular Vesicles (Manuscript Number: ijms-4023259).” We have carefully considered all comments and made the necessary revisions on the cover letter and the manuscript. The following provides detailed responses to each of the comments, with the reviewer’s comments presented in black font, our response in blue font, and all revisions in the manuscript are highlighted in red. Please see the details on attachment.
We believe these extensive revisions significantly enhancing the manuscript's originality, analytical rigor, and value to the field. We would like to express our great appreciation to you for your comments on our paper. Looking forward to hearing from you.
Thank you and best regards.
Sincerely yours.
Min Cheng, MD, Ph.D.
Department of Cardiology, Union Hospital
Tongji Medical College, Huazhong University of Science and Technology,
Wuhan 430022, P R China.
E-mail: min_cheng@hust.edu.cn

Round 2
Reviewer 1 Report
Comments and Suggestions for Authors
After the corrections have been made, the manuscript can be accepted for publication.
Author Response
Dear Reviewer,
We sincerely thank the reviewer for their positive evaluation and for acknowledging that the manuscript is now suitable for acceptance. We have retained all previous corrections and performed a final thorough proofreading to ensure clarity, consistency, and accuracy throughout the text.
Thank you and best regards.
Sincerely yours.
Min Cheng, MD, Ph.D.
Department of Cardiology, Union Hospital
Tongji Medical College, Huazhong University of Science and Technology,
Wuhan 430022, P R China.
E-mail: min_cheng@hust.edu.cn
Reviewer 2 Report
Comments and Suggestions for Authors
After thorough revisions, the value of the manuscript has increased. The authors focused on recent advances and expanded the clinical aspects of the topic, which is of great importance for both future theoretical research and practical applications.
Please also review the manuscript concerning the citation format. For example, line 95: “Trams et al10.” According to the journal’s requirements, citations should be placed in square brackets [ ]. This suggestion deals with the whole manuscript.
Author Response
Dear Reviewer,
We are very grateful for the reviewer’s encouraging remarks regarding the improved clinical and theoretical relevance of our manuscript entitled “Potential Biomarker and Therapeutic Tools for Pathological Cardiac Hypertrophy and Heart Failure: Extracellular Vesicles (Manuscript Number: ijms-4023259).” We are pleased that our efforts to highlight recent advances and strengthen the clinical perspective have been positively received.
Comment : Please also review the manuscript concerning the citation format. For example, line 95: “Trams et al10.” According to the journal’s requirements, citations should be placed in square brackets [ ]. This suggestion deals with the whole manuscript.
Response: We sincerely apologize for this oversight and thank the reviewer for bringing it to our attention. We have now carefully reviewed the entire manuscript and corrected all in-text citations to conform strictly to the journal’s formatting guidelines. Specifically, all numeric citations have been placed within square brackets (e.g., [10]) (Line 95). This formatting has been applied consistently across the entire text, including references to single citations, multiple citations, and citations integrated within sentences. We have also double-checked the reference list to ensure complete correspondence with the in-text citations and compliance with the journal’s style. The detailed changes are reflected in the revised manuscript.
We believe that all concerns raised by the reviewer have been adequately addressed in this revised version. The manuscript has been significantly strengthened through this revision process. We thank you again for your time and insightful comments on our paper. Looking forward to hearing from you.
Thank you and best regards.
Sincerely yours.
Min Cheng, MD, Ph.D.
Department of Cardiology, Union Hospital
Tongji Medical College, Huazhong University of Science and Technology,
Wuhan 430022, P R China.
E-mail: min_cheng@hust.edu.cn